# Boosting the catalysis of gold by $O_2$ activation at Au-SiO$_2$ interface

Yunlai Zhang[1,2,9], Junying Zhang[1,9], Bingsen Zhang [3], Rui Si[4], Bing Han[2,5], Feng Hong [1,2], Yiming Niu[2,3], Li Sun[2,5], Lin Li[5], Botao Qiao [5,6*], Keju Sun[7*], Jiahui Huang[1*] & Masatake Haruta[1,8]

Supported gold (Au) nanocatalysts have attracted extensive interests in the past decades because of their unique catalytic properties for a number of key chemical reactions, especially in (selective) oxidations. The activation of $O_2$ on Au nanocatalysts is crucial and remains a challenge because only small Au nanoparticles (NPs) can effectively activate $O_2$. This severely limits their practical application because Au NPs inevitably sinter into larger ones during reaction due to their low Taman temperature. Here we construct a Au-SiO$_2$ interface by depositing thin SiO$_2$ layer onto Au/TiO$_2$ and calcination at high temperatures and demonstrate that the interface can be not only highly sintering resistant but also extremely active for $O_2$ activation. This work provides insights into the catalysis of Au nanocatalysts and paves a way for the design and development of highly active supported Au catalysts with excellent thermal stability.

[1] Gold Catalysis Research Center, State Key Laboratory of Catalysis, Dalian Institute of Chemical Physics, Chinese Academy of Sciences, 116023 Dalian, China. [2] University of Chinese Academy of Sciences, 100049 Beijing, China. [3] Shenyang National Laboratory for Materials Science, Institute of Metal Research, Chinese Academy of Sciences, 110016 Shenyang, China. [4] Shanghai Synchrotron Radiation Facility, Zhangjiang Laboratory, 201204 Shanghai, China. [5] CAS Key Laboratory of Science and Technology on Applied Catalysis, Dalian Institute of Chemical Physics, Chinese Academy of Sciences, 116023 Dalian, China. [6] Dalian National Laboratory for Clean Energy, 116023 Dalian, China. [7] Key Laboratory of Applied Chemistry, College of Environmental and Chemical Engineering, Yanshan University, 438 Hebei Avenue, 066004 Qinhuangdao, China. [8] Research Center for Gold Chemistry and Department of Applied Chemistry, Graduate School of Urban Environmental Sciences, Tokyo Metropolitan University, Tokyo 192-0397, Japan. [9] These authors contributed equally: Yunlai Zhang, Junying Zhang *email: bqiao@dicp.ac.cn; kjsun@ysu.edu.cn; jiahuihuang@dicp.ac.cn

Catalysis by supported gold nanocatalysts has developed into one of the most important disciplines in heterogeneous catalysis field since the pioneering discoveries by Haruta[1,2] and Hutchings[3,4] that gold nanoparticles (NPs) can be very active heterogeneous catalysts. Supported gold nanocatalysts often offer unusual, and sometimes, unexpected catalytic performance for a number of key chemical reactions[5–10]. For example, oxide supported Au NPs can completely convert CO by molecular $O_2$ at ambient temperature or even lower, which is particularly applicable for CO cleaning in auto-emission elimination[11,12], respiratory mask[13,14] and $H_2$-fuel purification[15,16] and so on. In addition, supported Au nanocatalysts have much higher activity in water-gas shift (WGS) reaction than other catalyst formulas including the commercial catalysts in use[17,18]. Supported Au nanocatalysts are therefore highly promising and desired in industrial use[11,12,19,20], but their commercialization process has progressed very slowly which is in sharp contrast to the rapid expansion in fundamental studies. A major barrier is the low stability of supported Au catalysts, which stems from the fact that Au NPs are liable to sinter, through a Oswald ripening or other mechanism such as diffusion or coalescence of particles[10], and thus deactivate upon heating, during reaction or even under storage[21,22].

In the past 10 years, many efforts have been focused on addressing the sintering issue of supported Au NPs and significant progresses have been achieved. Strategies such as utilizing mesoporous materials to confine noble metal particles[23–25], using mixed or surface-modified oxide supports[26–28], developing new supports[29], coating the catalysts by inert oxide[30] and preparing the supported Au-based bimetallic alloy catalyst[31,32] can effectively improve the sintering resistance to 500 °C-calcination. However, it is still hard to realize good tolerance to temperatures above 600 °C due to the low Taman temperature of Au[33]. Very recently, a few catalysts have been reported to be sintering resistant to calcination in the temperature range of 600–800 °C[34–42] or even higher[43]. But this was achieved at the cost of losing their activity to different extent, thus resulting in a relatively higher temperature for CO complete elimination. The development of sintering resistant supported Au catalysts with high activity still remains a long-standing challenge.

For CO oxidation as well as many other oxidation reactions on supported Au catalysts, the $O_2$ activation is critically important[6,8,44,45]. Small Au NPs (e.g. <5 nm) are effective for $O_2$ activation, but the larger ones (>5 nm) are difficult to adsorb and activate $O_2$, and thus the activity decreases dramatically with the increase of Au NP size. Recently, we have discovered by theoretical calculation[46] that the formation of O–Au–O linear structure at Au-metal oxide interface can improve the activation of $O_2$ and then enhance CO oxidation activity. It was also found that the larger the electronegativity of the metal in metal oxides, the higher ability of this interface to activate $O_2$. According to this result we can expect a more active interface between Au and $SiO_2$ as $Si^{4+}$ in $SiO_2$ has much larger electronegativity compared with others (see Supplementary Table 1). However, the studies reported so far have not yet proven that $Au/SiO_2$ is highly active for CO oxidation, which may be due to the difficulty to build an appropriate Au-$SiO_2$ interface. Herein we successfully construct Au-$SiO_2$ interface by depositing $SiO_2$ thin layer onto Au/TiO_2 catalyst followed by high-temperature calcination. With this interface we reach an achievement that supported Au NPs (>5 nm) can realize CO 100% conversion at temperature below 0 °C and has a TOF of as high as 0.31 s$^{-1}$ at 25 °C even after 800 °C-calcination, similar to that of the standard Au/TiO_2 catalyst provided by Haruta Gold Inc. The catalyst exhibits not only extremely high activity but also excellent thermal stability and reaction durability, providing great opportunity for the practical application of gold catalyst. The finding in this work paves a way for the design and development of supported Au catalysts with good thermal stability and high activity.

## Results

**Structure characterization of Au@SiO$_2$/TiO$_2$ catalysts**. Au/TiO$_2$ catalyst was prepared by a deposition-precipitation (DP) method reported elsewhere[38] and calcined at 300 and 800 °C, denoted as Au/Ti-300 and Au/Ti-800 (Au/Ti-T) (Supplementary Fig. 1a), respectively. $SiO_2$ modification of Au/TiO$_2$ was performed by one pot co-deposition-precipitation of Au and Si precursors on TiO$_2$ support and also calcined at 300 and 800 °C, denoted as Au@SiO$_2$/Ti-300 and Au@SiO$_2$/Ti-800 (Au@SiO$_2$/Ti–T), respectively (Supplementary Fig. 1b). Details of the synthesis procedure are presented in Methods section.

Gold loadings of Au/Ti-fresh and Au@SiO$_2$/Ti-fresh were determined by ICP as 0.98 wt% and 0.93 wt% respectively, while the Si content was measured to be about 2.09 wt% for Au@SiO$_2$/Ti-fresh (Supplementary Table 2). The surface area measurements by $N_2$ adsorption show that Au@SiO$_2$/Ti–T catalysts have much higher BET surface area than Au/Ti–T catalysts, Supplementary Table 3. This must have originated from the doping of $SiO_2$ and the less sintered TiO$_2$ support compared with Au/Ti-800, which will be demonstrated later.

Figure 1 presents representative high-resolution transmission electron microscopy (HRTEM) images of various gold catalysts and the corresponding size distribution of Au NPs. It shows clearly that Au/Ti-300 has a narrow size distribution of gold NPs with a mean size of 2.5 nm, Fig. 1a. Au NPs were sintered seriously after calcination at 800 °C for only 2 h and a mean particle size of 20.4 nm was observed (Fig. 1b), in consistent with previous report[38]. The coating of $SiO_2$ species has negligible effect on the mean size of Au NPs with low-temperature calcination (300 °C) as Au@SiO$_2$/Ti-300 has similar mean Au size (3.5 nm) to that of Au/Ti-300 (Fig. 1c). The slightly larger size might come from the coverage of $SiO_2$ thin layer, Fig. 2. The presence of $SiO_2$ was confirmed by the Si–O–Si vibration by infrared measurements, Supplementary Fig. 2. For Au@SiO$_2$/Ti-800 sample the Au NPs were only aggregated slightly and still maintained a narrow size distribution with mean size of 6.4 nm (Fig. 1d), similar to previously reported Au/TiO$_2$-HAP-800 sample, where a strong metal-support interaction and a partial encapsulation of Au NPs existed[38]. More typical HRTEM images of all these four samples are shown in Supplementary Figs. 3–8.

The HRTEM characterization results were further confirmed by the XRD examinations. As can be seen in Supplementary Fig. 9, no Au diffraction patterns were observed on both Au/Ti-300 and Au@SiO$_2$/Ti-300, suggesting the highly dispersed Au species. After calcination at 800 °C, a peak assigned to the diffraction of Au (111) was observed on Au/Ti-800 while it is invisible on Au@SiO$_2$/Ti-800, indicating a much smaller Au particle size on the latter. The HRTEM and XRD characterization results demonstrated clearly the role of coating $SiO_2$ layer in preventing the sintering of Au NPs upon high-temperature calcination.

In addition, from the XRD patterns we can see that the crystal phase of TiO$_2$ support was transformed from P25 (a mixture of anatase and rutile) for Au/Ti-300 sample into pure rutile phase for Au/Ti-800 sample with 800 °C-calcination. However, phase transformation of TiO$_2$ on Au@SiO$_2$/Ti-800 was significantly inhibited by the introduction of $SiO_2$ on Au/TiO$_2$ (Supplementary Fig. 9). In addition, it shows that the -OH group on Au/TiO$_2$ sample disappeared after 800 °C-calcination while it remains almost unchanged after the introduction of $SiO_2$ layer, Supplementary Fig. 2, further confirming that the $SiO_2$ layer can

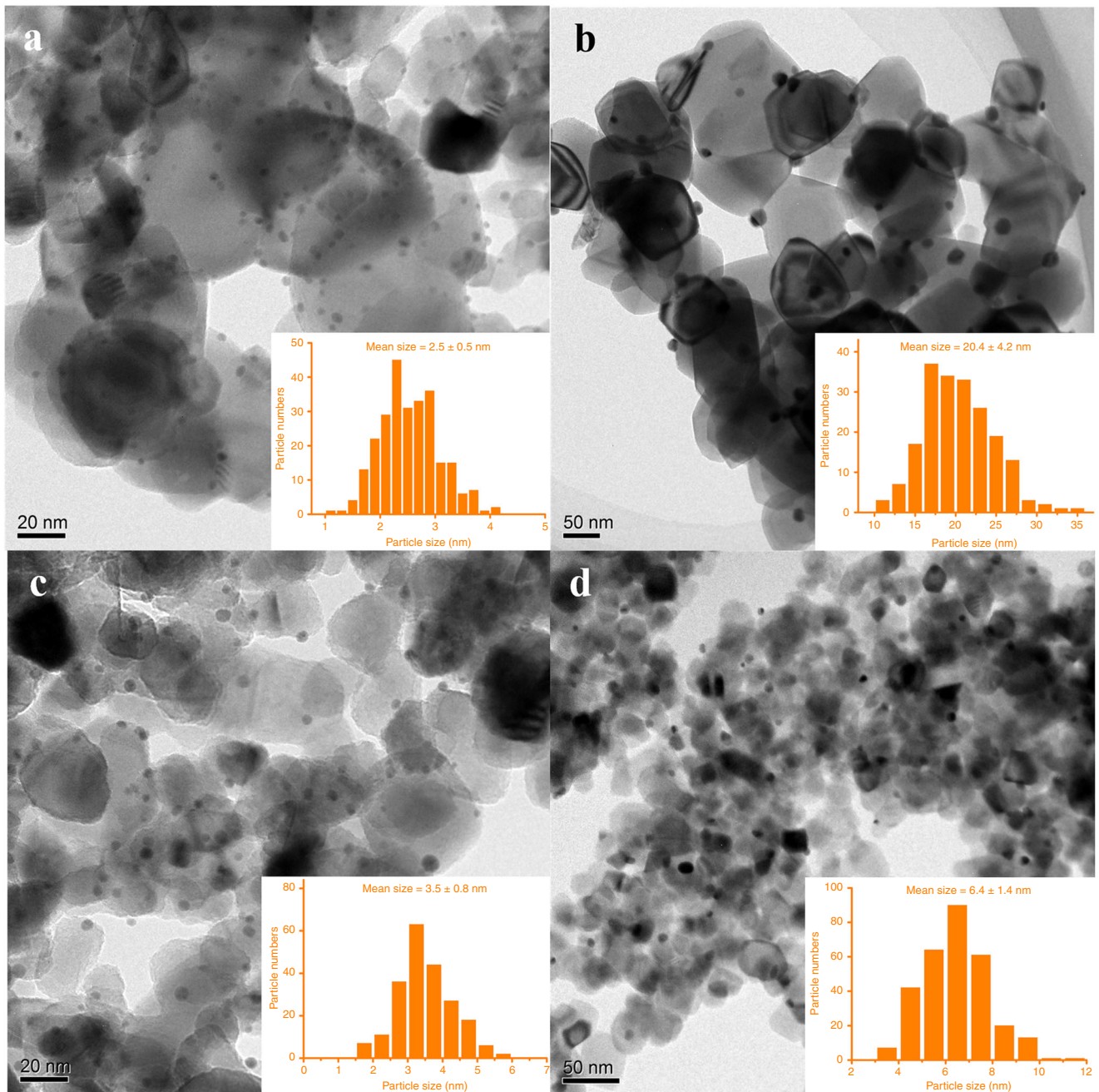

**Fig. 1 HRTEM images and corresponding size distributions of gold NPs for different gold catalysts. a** Au/Ti-300; **b** Au/Ti-800; **c** Au@SiO$_2$/Ti-300; **d** Au@SiO$_2$/Ti-800.

promote TiO$_2$ thermal stability. Obviously, the introduction of SiO$_2$ layer on the Au/TiO$_2$ greatly improves the thermal stability of not only Au NPs but also TiO$_2$ support. Both HRTEM and XRD characterizations unambiguously demonstrate that the SiO$_2$ coating over the surface of Au/TiO$_2$ can successfully prevent the agglomeration of gold particles under harsh heat treatment conditions.

We further employed STEM-EDS (Scanning Transmission Electron Microscopy-Energy Dispersive Spectrometer) elemental mapping to observe the silicon oxide distribution on Au@SiO$_2$/Ti–T samples, Supplementary Figs. 5 and 6. It revealed that Ti, O, Si atoms are superimposed and the Si element is distributed continuously and uniformly on Au@SiO$_2$/Ti-300, suggesting a homogeneous coating of SiO$_2$ layer over Au NPs and the TiO$_2$ support (Supplementary Fig. 5). Moreover, the elemental

mapping of a single gold particle on Au@SiO$_2$/Ti-300 confirmed that the surface of the gold particle is also covered by a thin SiO$_2$ film (Fig. 2e, f). Nevertheless, when the catalyst was annealed at 800 °C, the elemental mapping results exhibited that the SiO$_2$ film that covered the gold particle appears to aggregate to form island-like decoration (Fig. 2k, l). Meanwhile, it seems that the SiO$_2$ decoration dominantly located at the interfacial region between the TiO$_2$ support and gold particles (Fig. 2l), preventing the diffusion of Au atoms or the migration of Au NPs and thus inhibiting the sintering of gold NPs. HRTEM together with EDS mapping also verified that both TiO$_2$ support and gold particle are decorated by amorphous SiO$_2$ (Supplementary Figs. 5 and 6) with a thickness of a few atom layers (Supplementary Figs. 7 and 8). Furthermore, a detailed analysis of the HRTEM images found that most of the gold particles became partially coated by the

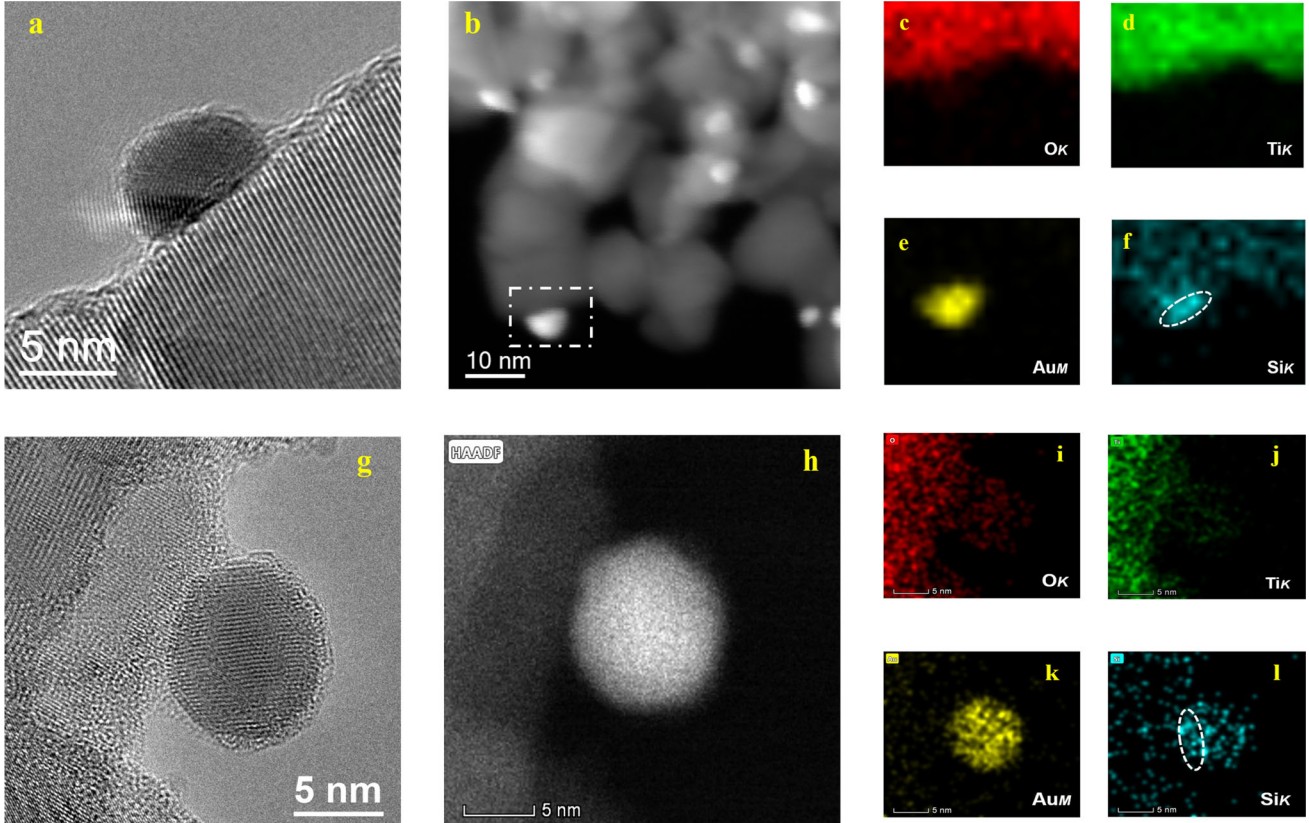

**Fig. 2 HRTEM and HAADF-STEM images and EDS mapping of different catalysts. a–f** Au@SiO$_2$/Ti-300; **g–l** Au@SiO$_2$/Ti-800.

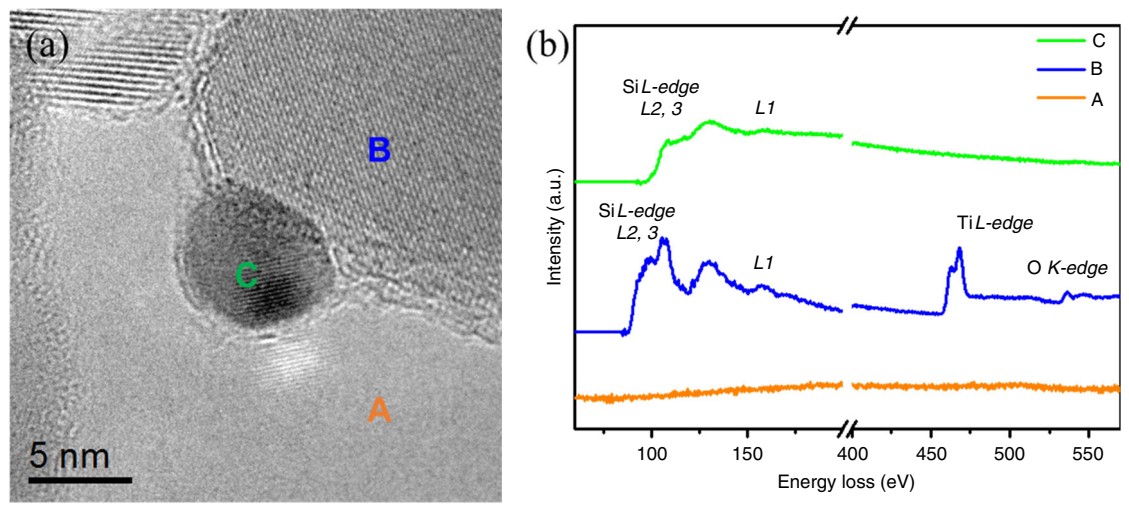

**Fig. 3 The EELS analysis of Au@SiO2/Ti-800. a** TEM image; **b** EELS spectra.

amorphous SiO$_2$ as the calcination temperature increased from 300 to 800 °C (Supplementary Figs. 7 and 8). Obviously, the thermal annealing process induced the assemblage of SiO$_2$ towards the interface of TiO$_2$ and gold particles which will play a vital role in the anti-sintering performance of gold catalysts.

Electron energy loss spectroscopy (EELS) examination was also performed to detect the SiO$_2$ cover layer and their chemical states (Fig. 3). Si on TiO$_2$ support (region B, Fig. 3a) exhibited Si–O$_4$ tetrahedral structure, suggesting a SiO$_2$ structure, whereas Si on Au NPs (region C, Fig. 3a) exhibited more metal character (Fig. 3b), implying an electron transfer from Au to SiO$_2$ which

is most probably aroused from a strong interaction between Au and SiO$_2$.

**Electronic state of Au@SiO$_2$/TiO$_2$ catalyst.** XPS measurements were performed to study the electronic state of Au as well as the interaction between Au and supports. As shown in Fig. 4a, on the fresh Au/TiO$_2$ sample without calcination treatment and Au/Ti-300 the Au 4$f_{7/2}$ spectra were broad and can be fitted into two peaks at about 83.8 eV and 83.2 eV, respectively, with a ratio of the former to the latter being 0.27 and 0.22, respectively

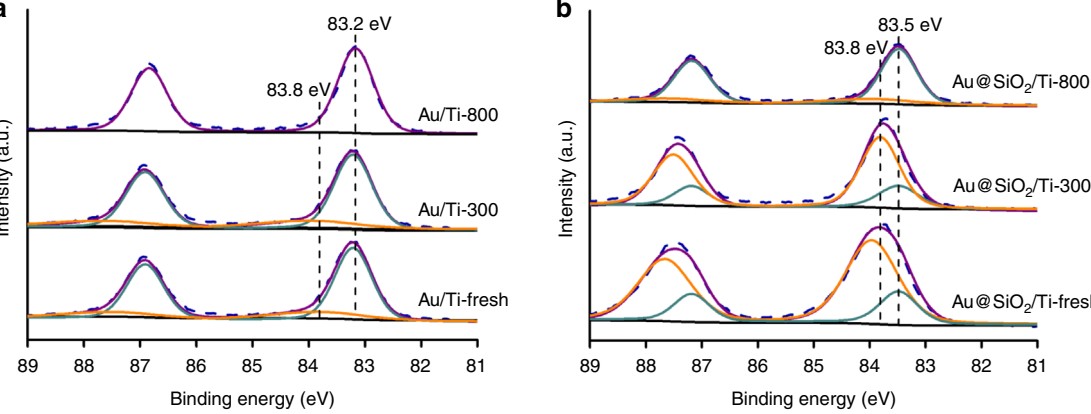

**Fig. 4 Au 4$f$ XPS spectra of various catalysts. a** Au/Ti-fresh, Au/Ti-300 and Au/Ti-800; **b** Au@SiO$_2$/Ti-fresh, Au@SiO$_2$/Ti-300, and Au@SiO$_2$/Ti-800.

(Supplementary Table 4). This suggests that a small part of Au species are positively charged in fresh catalyst and Au/Ti-300, which is as expected and consistent to previous studies[47–49]. After calcined at 800 °C, the positively charged Au decomposed completely into metallic Au with a 4$f_{7/2}$ binding energy of 83.2 eV on Au/Ti-800, also consistent with previous studies[47,50]. After the introduction of SiO$_2$, the electronic behaviors of Au species changed significantly. Over Au@SiO$_2$/Ti-fresh sample without any heat-treatment Au species also existed as the mixture of positively charged state and metallic state, but with a much higher ratio of the former to the latter (3.76, Supplementary Table 4). After calcination at 300 °C the situation didn't change much, only the ratio of positively charged Au decreased slightly (from 3.76 to 3.0). This suggests a much stronger interaction between Au and SiO$_2$–TiO$_2$ support which prevents the complete decomposition of positively charged Au species. Of more interest, even after calcination at 800 °C the binding energy of Au 4$f_{7/2}$ (83.5 eV) is slightly higher than that on Au/Ti samples (83.2 eV) (Fig. 4b). It seems incredible that the Au@SiO$_2$/Ti-300 contains such high proportion of positively charged Au species after being calcined at 300 °C. To further confirm the chemical state of Au species, CO adsorption diffuse reflectance infrared Fourier transform (CO-DRIFT) spectra were measured. To avoid the possible Au reduction during CO adsorption[44], all the spectra were collected at a very low temperature (−150 °C), Supplementary Fig. 10. It showed that over Au/Ti-300 CO adsorption on Au sites appeared at 2103 cm$^{-1}$, suggesting a metallic state. However, on Au@SiO$_2$/Ti-300 sample, in addition to a slightly blue-shifted CO adsorption at 2104 cm$^{-1}$, a shoulder peak centered at ~2123 cm$^{-1}$ was observed, unambiguously demonstrating the presence of positively charged Au species. Therefore, combining the EELS, XPS and CO-DRIFT results, we can safely conclude that there existed positively charged Au species on Au@SiO$_2$/TiO$_2$ samples after calcination at 300 and even 800 °C which originated from a strong interaction between Au and SiO$_2$ layer.

In addition, the intensity of CO adsorption peaks was compared, which to some extent reflects the amount of exposed surface sites. As expected, Au/Ti-300 sample has the most intense CO adsorption on both Au and TiO$_2$ surface (2176 cm$^{-1}$). However, for Au/Ti-800 sample the CO adsorption on TiO$_2$ surface decreased significantly due to the decrease of surface area, and that on Au sites disappeared totally probably due to the very weak adsorption of CO on large Au particles[38]. On the other hand, for Au@SiO$_2$/Ti-300 CO adsorption on both TiO$_2$ surface and Au decreased dramatically, evidencing a coating effect of SiO$_2$. Interestingly, after calcined at 800 °C, the CO adsorption on TiO$_2$ increased, while that on Au sites decreased further. The former is in well agreement with the shrink of SiO$_2$ layer during

calcination and the latter is due to the aggregation of gold particles.

**Catalytic performance**. The catalytic performance of Au/TiO$_2$ and Au@SiO$_2$/TiO$_2$ series samples for CO oxidation were tested in a gas composition of 1 vol.% CO + 20 vol.% O$_2$ + 79 vol.% N$_2$, with a space velocity of 20 L g$_{cat.}^{-1}$ h$^{-1}$. As shown in Fig. 5a, the CO oxidation activity of Au/TiO$_2$ catalysts depends strongly on the calcination temperature: Au/Ti-300 with a mean Au size of 2.5 nm exhibits a very high CO oxidation activity that 100% conversion can be achieved at −30 °C. However, after calcination at 800 °C the Au/Ti-800 catalyst with a much larger mean Au size of 20.4 nm shows dramatically decreased catalytic activity, and the CO conversion at 150 °C is only 75%. This result demonstrates clearly that the activity for CO oxidation strongly depends on the size of Au NPs, in consistent with previous observations on Au/TiO$_2$ catalysts[51–54]. Interestingly, the Au@SiO$_2$/TiO$_2$ series catalysts show distinct different catalytic performance. The Au@SiO$_2$/Ti-300 with an average Au size of 3.5 nm shows a much lower activity compared with Au/Ti-300 and CO total conversion is realized at 100 °C, Fig. 5a. The much lower activity might be due to the coverage of Au NPs by the SiO$_2$ layer. However, Au@SiO$_2$/Ti-800 with an average Au size of 6.4 nm exhibits a drastically improved activity that CO total conversion can be realized at 0 °C. This is surprising because so far no supported Au catalysts with calcination temperature ≥600 °C can realize 100% CO conversion at ambient temperature. In order to obtain the intrinsic activity of Au@SiO$_2$/Ti-800 to compare with those of supported Au catalysts reported in literatures, we measured its specific rate at 25 °C and calculated the corresponding turnover frequency (TOF). As shown in Supplementary Table 5, it yields a TOF of as high as 0.31 s$^{-1}$, almost same to the two standard Au/TiO$_2$ catalysts provided by Haruta Gold Inc (1.0 wt% Au/TiO$_2$, RR2Ti) and World Gold Council (1.47 wt% Au/TiO$_2$), respectively, and ~2–15 times higher than that of the most active Au catalysts with calcination at temperatures ≥600 °C reported so far, evidencing the extremely high activity of Au@SiO$_2$/Ti-800 sample.

The sintering of Au NPs is the main reason for the deactivation of supported Au catalyst for CO oxidation, especially at high temperatures[38,55]. Since Au@SiO$_2$/Ti-800 showed good sintering resistance, a good stability during reaction can be reasonably expected. This was verified by a CO oxidation stability test at 400 °C, where an initial conversion lower than 100% was tuned to avoid the activity saturation by increasing the gas hourly space velocity, Fig. 5b. It turns out that except an initial fluctuation no deactivation was observed during 100-hour test. The recycle

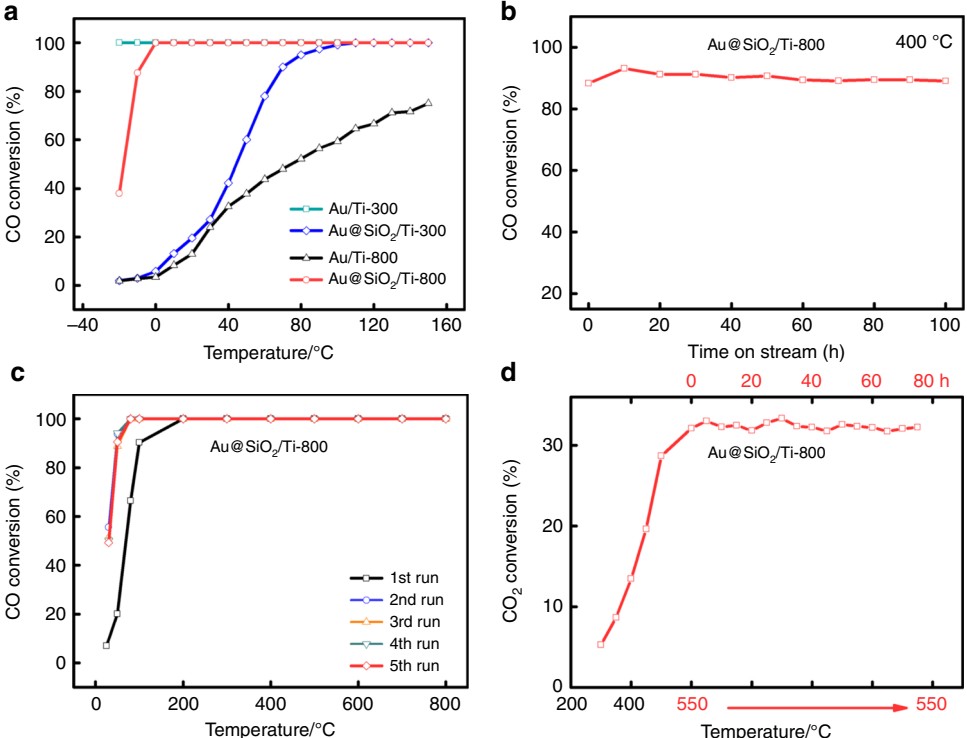

**Fig. 5 The catalytic performance of gold catalysts in various reactions. a** CO conversion as a function of reaction temperature on Au/Ti-T and Au@SiO$_2$/Ti-T; **b** CO conversion as a function of reaction time on Au@SiO$_2$/Ti-800 catalyst tested at 400 °C; **c** CO conversion as a function of reaction temperature on Au@SiO$_2$/Ti-800 catalyst for different cycles. Gas flow: 1 vol.% CO + 20 vol.% O$_2$ + 79 vol.% N$_2$, and the space velocity for **a**, **b**, and **c** is 20 L g$_{cat.}^{-1}$ h$^{-1}$, 1005 L g$_{cat.}^{-1}$ h$^{-1}$ and 60 L g$_{cat.}^{-1}$ h$^{-1}$, respectively. **d** The catalytic performance for RWGS reaction on Au@SiO$_2$/Ti-800. Gas flow: 10 vol.% CO$_2$ + 90 vol.% H$_2$, and the space velocity is 15 L g$_{cat.}^{-1}$ h$^{-1}$.

performance of this catalyst was also examined since in practical applications (especially for auto-exhaust elimination) catalysts often need to go through cold-hot cycling frequently. As shown in Fig. 5c, in five cycles running up to 800 °C the catalyst activity kept unchanged except an increase in second run, suggesting the excellent stability and cyclicity in practical applications. The increased activity in second run implies that the catalyst can be further activated upon high-temperature reaction, which may be due to a subtle change of the metal support interaction during reaction[56].

Water is almost unavoidable in any heterogeneous catalysis system which is usually thought to facilitate the sintering of metal particles upon heating treatment thus harmful to the catalyst durability. Especially, the low stability of SiO$_2$ against the hydrothermal condition may arouse concerns. To evaluate the stability of Au@SiO$_2$/Ti-800 in the presence of water, reverse water gas shift (RWGS), a typical high-temperature reaction that has attracted increasing interest in recent years due to its importance in CO$_2$ activation and utilization[57,58], was further studied. As shown in Fig. 5d, at 550 °C the CO$_2$ conversion on our Au@SiO$_2$/Ti-800 sample is about 30%, corresponding to a generation of 3 vol% of H$_2$O in the reaction gas. Our catalyst can run stably at 550 °C for at least 75 h without any deactivation, demonstrating an excellent sintering resistance to the presence of water.

**Identifying the origin of the high activity.** The HRTEM, EELS, and XRD characterization results unambiguously demonstrated that the introduction of a SiO$_2$ cover layer can inhibit the sintering of Au NPs upon high-temperature calcination up to 800 °C. This is not very strange since the role of physical coverage has been

identified and been widely used to fabricate sintering-resistant Au catalysts[59–61]. However, it is quite striking that the catalyst has such a high activity. As our previous Au/TiO$_2$-HAP-800 catalyst with a similar Au NP size (~6 nm) did not show such a high activity (where the activity originated from the interface of Au–TiO$_2$)[38], the improved activity in this catalyst must have come from the formation of Au–SiO$_2$ interface. This seems incredible since it is well-accepted that TiO$_2$ is an active support for CO oxidation whereas SiO$_2$ is an inert one[62]. To confirm the critical role of the Au-SiO$_2$ interface we performed a series of characterizations and control experiments. We firstly examined the Au@SiO$_2$/Ti-800 sample with an aberration corrected STEM (AC-STEM) to detect the possible residual small Au NPs/clusters. The images revealed no existence of such small Au aggregates (Supplementary Fig. 11). X-ray adsorption measurement and the corresponding fitting results showed that the coordination number (CN) of Au–Au for Au@SiO$_2$/Ti-800 is smaller than that for Au/Ti-800 while larger than others (Supplementary Table 6 and Supplementary Fig. 12), suggesting that Au@SiO$_2$/Ti-800 do have larger Au NPs than other samples except Au/Ti-800. These characterization results exclude the possibility that the high activity may originate from the residual very small Au NPs/clusters. Another possibility is that the introduction of SiO$_2$ may create some very defective TiO$_2$ species, which can greatly improve the CO oxidation activity. In order to explore this possibility, an electron paramagnetic resonance (EPR) examination was further performed. The results showed that in Au/Ti-300 a very weak peak at $g = 1.975$ was observed which can be ascribed to the Ti$^{3+}$ (implying an existence of O vacancy), Supplementary Fig. 13. However, on Au@SiO$_2$/Ti-800 this peak is completely invisible, suggesting no O vacancy. This possibility is therefore excluded as well. Yates group recently demonstrated a new CO

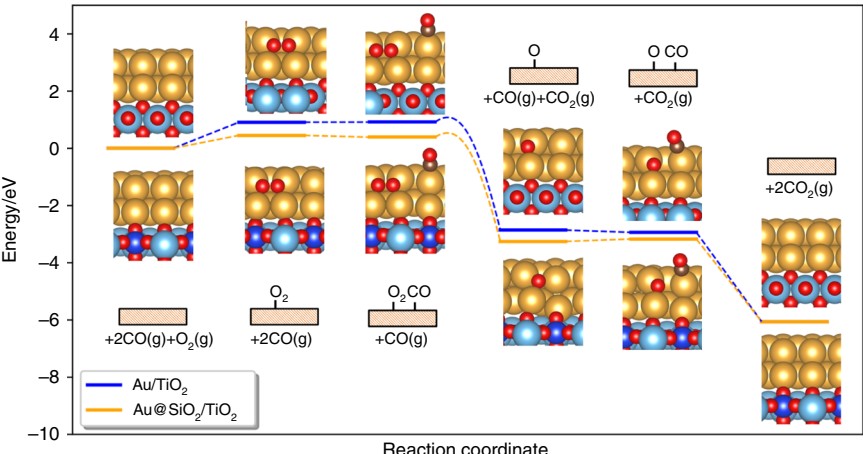

**Fig. 6 Calculated reaction mechanism of CO oxidation on Au/TiO₂ and Au@SiO₂/TiO₂.** The structures with only adsorbed species are shown in the inserted figures and the gaseous species are ignored. The gold, red, cyan, brown, and blue balls are Au, O, Ti, C, and Si atoms, respectively.

oxidation mechanism on Au/TiO₂ by low-temperature DRIFT spectroscopy[63], which has been developed into a new method to examine the interfaces between the metal NPs and the TiO₂ support[38,64]. We therefore performed a similar measurement at −150 °C and −100 °C, respectively. In both cases after the introduction of gas mixture of CO and air, no CO₂ generation was observed at all, suggesting nonexistence of such a highly active Au–TiO₂ interface in our Au@SiO₂/Ti-800 sample (Supplementary Fig. 14). It is well-known that the residual chlorides have significant effect on the activity of Au[65]. We therefore measured the chloride content by using an ion chromatography, Supplementary Table 7. It shows that on both catalysts the chloride contents are extremely low (≤3 ppm), only about 0.03% of that of Au amount. We hence do not think that the high activity of Au@SiO₂/Ti-800 stemmed from a low concentration of halide.

By excluding the above possibilities, the most possible explanation is the formation of active Au–SiO₂ interface. To provide direct evidence, we etched the SiO₂ layer with NaOH from Au@SiO₂/Ti-800 (denoted as Au@SiO₂/Ti-800-NaOH). The success removal of SiO₂ was evidenced by the disappearance of Si–O–Si vibration in IR spectrum (Supplementary Fig. 15a). A measurement of the Si content in the etched sample shows that most SiO₂ (~93%) had been removed (Supplementary Table 2). Catalytic performance test reveals that the activity decreased drastically after the removal of SiO₂ cover layer (Supplementary Fig. 15b). One may argue that the decreased activity may stem from the effect of NaOH treatment rather than the removal of the SiO₂ despite the fact that NaOH treatment generally improves the activity[44,66]. To exclude this possibility, we further prepared an Au/TiO₂ with similar Au NP size (~6.7 nm, Supplementary Fig. 16) by carefully calcining the sample at 600 °C for 3 h. It is also much less active than Au@SiO₂/Ti-800 catalyst but shows a similar activity to the etched sample of Au@SiO₂/Ti-800-NaOH, Supplementary Fig. 17. This unambiguously demonstrates that the lower activity of Au@SiO₂/Ti-800-NaOH originated from the lack of Au–SiO₂ interface rather than the influence of NaOH treatment. The change of active sites should result in change in active energy. The experimentally measured apparent activation energies are indeed totally different (26.1 kJ mol⁻¹ for Au@SiO₂/Ti-800 and 37.4 kJ mol⁻¹ for Au@SiO₂/Ti-800-NaOH, Supplementary Fig. 18a) and the values are very similar to the theoretically calculated ones (20.5 vs 43.1 kJ mol⁻¹), Supplementary Fig. 18b).

Considering that the lattice O of SiO₂ is hard to be extracted, the reaction according to a redox mechanism would be therefore

less active on Au@SiO₂/Ti-800 than on Au@SiO₂/Ti-800-NaOH. Water gas shift reaction is suggested to proceed mainly according to a redox mechanism at elevated temperature[67]. We therefore compared the catalytic performance of Au@SiO₂/Ti-800 before and after the etching of SiO₂ in this reaction. The result presented in Supplementary Fig. 19 shows clearly that at elevated temperature the activity is obviously higher after SiO₂ being etched, which is in consistent with the prediction but in contrast to CO oxidation.

The less reductive lattice O of Au@SiO₂-Ti-800 was further confirmed by hydrogen temperature-programmed reduction (H₂-TPR) measurements. A very weak surface O₂ reduction peak at 300 °C was present on Au/Ti-300, while no reduction peak was observed on Au@SiO₂/Ti-800, Supplementary Fig. 20. A CO pulse reaction was further performed to investigate the activity of the lattice oxygen (Supplementary Fig. 21). On both Au@SiO₂/Ti-800 and Au/Ti-300 catalysts, CO₂ product is detected but the intensity on Au@SiO₂/Ti-800 is much lower, corroborating the above conclusion that the introduction of SiO₂ in fact inhibits the reducibility of TiO₂. In addition, the negligible CO₂ product in both cases suggests that the Mars-van-Krevelen (MvK) mechanism does not dominate on both samples, at least at ambient temperature. This agrees well with the previous reports that MvK mechanism only dominates on very reducible support such as FeOₓ[44,68] and the fact that on Au/TiO₂ catalyst MvK mechanism only became prominent at reaction temperature >80 °C[69,70].

**DFT calculation.** To better understand the unique role of Au–SiO₂ interface in promoting activity of CO oxidation, the detailed reaction mechanisms as well as reaction rates were calculated by micro-kinetic studies based on DFT calculation results. The whole reaction processes can be described in Fig. 6 and Supplementary Fig. 22. The sites to activate O₂ are located on Au (100) facets between first and second layer near the interface of Au/TiO₂ or Au@SiO₂/TiO₂, and the CO molecule adsorbs on the corner site of Au clusters near the adsorbed O₂ molecule. The adsorbed O₂ molecule dissociates to form two oxygen atoms with the help of double linear O-Au-O structure at the perimeter of Au/TiO₂ or Au@SiO₂/TiO₂, and then two CO molecules react with two oxygen atoms to generate two CO₂ molecules, respectively. It can be found that the adsorption free energy of O₂ on Au/TiO₂ is 0.91 eV, much weaker than that of 0.45 eV on Au@SiO₂/TiO₂, which indicates more adsorbed O₂ molecules on Au@SiO₂/TiO₂. The relatively stronger adsorption energy of

$O_2$ on $Au@SiO_2/TiO_2$ can be ascribed to the $Si^{4+}$ sites which attract more electrons from anti-bonding orbitals of O-Au-O structure compared to $Ti^{4+}$ sites as shown in Supplementary Fig. 23. In addition, the dissociation barrier for $O_2$ of 0.12 eV on $Au/TiO_2$ is higher than that of 0.09 eV on $Au@SiO_2/TiO_2$ (Supplementary Fig. 24), implying an easier $O_2$ activation on $Au@SiO_2/TiO_2$. The overall reaction steps of CO oxidation on $Au/TiO_2$ and $Au@SiO_2/TiO_2$ catalysts are shown in Supplementary Tables 8 and 9. The $X_{RC}$ (degree of thermodynamic rate control) values of the $O_2$ dissociation on both of $Au/TiO_2$ and $Au@SiO_2/TiO_2$ are near 1.00, which indicates that the rate-determining step (RDS) for CO oxidation is $O_2$ activation. The coverage of $O_2$ and O can be quantified as $1.49 \times 10^{-17}$ and $7.43 \times 10^{-20}$ on $Au/TiO_2$, and $2.10 \times 10^{-10}$, and $1.25 \times 10^{-13}$ on $Au@SiO_2/TiO_2$, respectively. It can be found that the coverage of $O_2$ and O are increased significantly on $Au@SiO_2/TiO_2$ compared to $Au/TiO_2$, indicating the coexistence of $SiO_2$ and $TiO_2$ could strengthen the adsorption of oxygen species on gold clusters. The calculated reaction rate of CO oxidation on $Au/TiO_2$ is only $3.01 \times 10^{-7} \, s^{-1} \, site^{-1}$, which is six orders of magnitude lower than that of $0.702 \, s^{-1} \, site^{-1}$ on $Au@SiO_2/TiO_2$. It clearly shows that the coexistence of $SiO_2$ and $TiO_2$ could promote the CO oxidation on $Au/TiO_2$ system, and the promotion effects can be contributed to the enhancement of $O_2$ adsorption and activation by the formed Au–$SiO_2$ interface.

In summary, we have fabricated a $SiO_2$ decorated $Au/TiO_2$ catalyst by covering $SiO_2$ islands onto the Au NP surface to form a more active Au–$SiO_2$ interface. This catalyst is not only thermally stable but also highly active for CO oxidation to realize CO total conversion at 0 °C even after calcination at temperature as high as 800 °C. Experiments together with computational studies revealed that the formation of Au–$SiO_2$ interface is critical for the high activity which may open up a window for the development of highly active and stable supported metal catalysts for CO oxidation.

## Methods

**Raw materials**. Sodium hydroxide (NaOH, 98%), P25 ($TiO_2$) was purchased from Evonik Degussa. Hydrogen terachloroaurate (IV) hydrate ($HAuCl_4 \cdot 4H_2O$) and tetraethoxysilane (TEOS) were purchased from Acros (ACS reagent). All the reagents were used as received without further treatments.

**Preparation of gold samples**. $Au/TiO_2$ catalysts were prepared by a deposition-precipitation (DP) method. In a typical procedure, a certain amount of $HAuCl_4$ was dissolved in 100 mL water to achieve a $HAuCl_4$ aqueous solution with the concentration of 0.73 mmol $L^{-1}$. Then, NaOH solution (1 mol $L^{-1}$) was added to adjust the pH of $HAuCl_4$ solution to 8. After that, 1.17 g $TiO_2$ (P25) support was added under vigorous stirring. The mixture was heated to 70 °C and kept for 3 h during which the pH value was maintained at 8 with NaOH solution (0.1 mol $L^{-1}$). Finally, the resulting precipitate was filtered and washed with 2 L deionized water, then dried at 80 °C in oven over night (denoted as Au/Ti-fresh). The as-prepared $Au/TiO_2$ was divided to three parts and calcined at 300 and 800 °C in air for 2 h, and at 600 °C in air for 3 h, respectively, denoted as Au/Ti-T ($T$ = 300 and 800 °C) and Au/Ti-600-3h. The $SiO_2$ modified $Au/TiO_2$ catalysts were prepared in a same procedure except that TEOS was added to the mixture of $TiO_2$ and $HAuCl_4$ solution after the suspension was heated to 70 °C. The obtained $SiO_2$ modified samples were calcined at 300 and 800 °C in air for 2 h, respectively, denoted as $Au@SiO_2$/Ti-T ($T$ = 300 and 800 °C). In all, 100 mg of $Au@SiO_2$/Ti-800 was further etched by aqueous NaOH solution (pH = 12) at 70 °C for 3 h under vigorous stirring, and then filtered and washed using deionized water until the pH of filtrate was neutral. The obtained catalyst was dried at 80 °C in oven over night and denoted as $Au@SiO_2$/Ti-800-NaOH.

**Characterization**. The actual loadings of Au and Si were determined using an inductively coupled plasma atomic emission spectrometer (ICP-AES) on the Plasma-Spec-Ii instrument (Leeman Hydraulic Technology Co., Ltd.).

High resolution transmission electron microscopy (HRTEM) and high-angle annular dark-field scan transmission electron microscopy (HAADF-STEM) images were obtained on JEOL-2100F with an accelerative voltage of 200 kV. The aberration corrected HAADF-STEM examination was performed on a JEM-ARM300F electron microscope with an accelerative voltage of 300 kV, which

guarantees a resolution of 0.08 nm. Samples were dispersed in ethanol by ultrasonication, and the resulting solution dropped on to carbon films supported on copper grids.

X-ray diffraction (XRD) analysis was performed with a D/Max-2500/PC diffractometer (Rigaku) equipped with a Cu Kα radiation source ($\lambda = 0.15432$ nm), and with an operation voltage and operation current of 40 kV and 40 mA, respectively.

X-ray photoelectron spectroscopy (XPS) were determined by XPS on an ESCALAB250 X-ray photoelectron spectrometer and using monochromatic Al Kα radiation (ESCALAB250, thermo VG), and the binding energies of samples were calibrated by taking the carbon 1 s peak as reference (284.6 eV).

Temperature-programmed reduction by $H_2$ ($H_2$-TPR) was carried out on an Auto Chem II 2920 automatic catalyst characterization system. Typically, 100 mg of the sample was loaded into a U-shape quartz reactor and purged with He at 393 K for 2 h. Then, the reactor was cooled to 293 K and the flowing gas was switched to a 10 vol% $H_2$/He. The catalyst was heated to 873 K at a ramping rate of 10 K $min^{-1}$ and the TCD signal was recorded.

The CO pulse reaction was also performed on an Auto Chem II 2920 automatic system. Typically, 100 mg of the sample was loaded into a U-shape quartz reactor and purged with He at 393 K for 2 h. Then, the reactor was cooled to 323 K and kept under the flow of He and 10 vol% CO/He was pulsed into the reactor every five minutes during which process the MS signal was recorded.

Electron paramagnetic resonance (EPR) measurements were performed at 100 K using a Bruker EPR A200 spectrometer with a 100-kHz magnetic field modulation at a microwave power of 20.0 mW. The magnetic field was calibrated using 5,5-dimethyl-1-pyrroline N-oxide (DMPO) as a standard.

Fourier transform Infrared (FT-IR) spectra were collected on a Bruker VERTEX 70 FT-IR spectrometer scaled at 4000 $cm^{-1}$ to 640 $cm^{-1}$ with a resolution of 4 $cm^{-1}$.

CO adsorption IR spectra on various catalysts were recorded on a Bruker VERTEX 70 FT-IR spectrometer equipped with an MCT detector (Harrick Scientific Products, Inc.). The catalyst was firstly pretreated at 393 K with He (30 mL $min^{-1}$) for 0.5 h, and was cooled down to 123–173 K. Then, the background spectra were recorded, and adsorbed gas was introduced into the reaction cell to desirable pressures via a leak valve and the IR spectra were recorded.

The X-ray absorption fine structure (XAFS) spectra at Au $L_3$ ($E_0 = 11919$ eV) edge was performed at BL14W1 beamline of Shanghai Synchrotron Radiation Facility (SSRF) operated at 3.5 GeV under "top-up" mode with a constant current of 250 mA. The XAFS data were recorded under fluorescence mode with a Lytle-type ion chamber. The energy was calibrated accordingly to the absorption edge of pure Au foil. Athena and Artemis codes were used to extract the data and fit the profiles. For the extended X-ray absorption fine structure (EXAFS) part, the Fourier transformed (FT) data in $R$ space were analyzed by applying first-shell approximate and metallic Au model for Au–O and Au–Au contributions, respectively. The passive electron factors, $S_0^2$, were determined by fitting the experimental data on Au foils and fixing the coordination number (CN) of Au–Au to be 12, and then fixed for further analysis of the measured samples. The parameters describing the electronic properties (e.g., correction to the photoelectron energy origin, $E_0$) and local structure environment including $CN$, bond distance ($R$) and Debye-Waller factor ($\sigma^2$) around the absorbing atoms were allowed to vary during the fit process. The fitted ranges for $k$ and $R$ spaces were selected to be $k = 3$–10 $Å^{-1}$ with $R = 1$–3 Å ($k^3$ weighted).

The $Cl^-$ ion concentration was tested at 30 °C using a dionex ion chromatography instrument (ICS1100). In all, 100 mg of different catalysts were taken and soaked in 30 ml 0.5 wt% ammonia solution under strong agitation for 12 h. After filtration through a 0.22 μm PTFE filter, the liquid samples were analyzed at a dionex ion chromatography (ICS1100).

DFT calculations are carried to investigate the CO oxidation reaction on Au/$TiO_2$ and $Au@SiO_2/TiO_2$ by using Vienna Ab Initio Simulation Package (VASP). The exchange-correlation energy and potential are described by the generalized gradient approximation (GGA) in the form of Perdew-Burke-Ernzerhof (PBE). A cutoff of 400 eV is used for projector augmented wave (PAW) potentials. Since the promotion effect of added $SiO_2$ on CO oxidation is significant only after 800 °C calcination, we consider that some $SiO_2$ enter $TiO_2$ lattice and replace some $TiO_2$ near the interface of $Au/TiO_2$. Therefore, the models of $Au/TiO_2$ (110) and $Au@SiO_2/TiO_2$ (110) with some interfacial $TiO_2$ substituted by $SiO_2$ are built as shown in Supplementary Fig. 21. A $2 \times 4$ supercell is used for $TiO_2$ (110) three-trilayer slab and $2 \times 2 \times 1$ k-point mesh is adopted for $Au/TiO_2$(110) and $Au@SiO_2/TiO_2$(110). The structures are optimized and two bottommost trilayers of $TiO_2$ (110) slabs are fixed at their bulk positions. The free energy instead of reaction heat is involved, and the contributions of vibration are ignored in both of gaseous and adsorbed states due to the variety of the energy differences induced by the contribution of vibration is slight. To calculate the free energy, the temperature is used as 273.15 K and the partial pressures of CO, $O_2$, and $CO_2$ are 1000 Pa, 20000 Pa, and 1000 Pa, respectively. A micro-kinetic model is employed to quantitatively calculate the activity of CO oxidation reaction and the details are shown in supporting information and elsewhere[71].

**Catalytic performance test**. The catalytic activity of CO oxidation was evaluated with a fixed bed reactor at atmospheric pressure. Typically, 150 mg of catalyst was equipped in the reactor, and a mixture of 1 vol.% CO, 20 vol.% $O_2$, and $N_2$ balance

was passed over with a flow rate of 50 mL min$^{-1}$ (space velocity of 20 L g$_{cat.}$$^{-1}$ h$^{-1}$). And the temperature was increased from $-20\,°C$ to $150\,°C$ with a rate of $2\,°C$ min$^{-1}$. The long-term stability test of Au@SiO$_2$/Ti-800 at $400\,°C$ was performed in the same reactor and gas mixture, while the catalysts was used only 5 mg and the gas flow was controlled as 83.7 mL min$^{-1}$ (space velocity of 1005 L g$_{cat.}$$^{-1}$ h$^{-1}$), in order to keep the CO conversion below 100%. The high-temperature stability test of Au@SiO$_2$/Ti-800 was also carried out under a mixture of 1 vol.% CO, 20 vol.% O$_2$, and N$_2$ balance, with the catalyst of 50 mg, and the gas flow of 50 mL min$^{-1}$ (space velocity of 60 L g$_{cat.}$$^{-1}$ h$^{-1}$). The temperature was increased from room temperature to $800\,°C$ with a rate of $10\,°C$ min$^{-1}$. When the temperature reached $800\,°C$, the gas was switched to N$_2$, and cooled down to room temperature. Then, the second run was started. For the catalytic performance of Au@SiO$_2$/Ti-800 in RWGS (reverse water gas shift) reaction, the temperature was increased from 300 to $550\,°C$ with a rate of $5\,°C$ min$^{-1}$ and maintained at $550\,°C$ for 75 h. And the gas was a mixture of CO$_2$ and H$_2$ with a ratio of 1:9. The total flow and catalyst mass was kept as 50 mL min$^{-1}$ and 200 mg, respectively (space velocity of 15 L g$_{cat.}$$^{-1}$ h$^{-1}$). The inlet and outlet gas compositions were analyzed online by a gas chromatograph (Agilent-GC7890B) equipped with a 5A sieve column and a flame ionization detector (FID) where the CO and CO$_2$ were totally converted into methane before flowing into the detector. And the water gas shift activity of Au@SiO$_2$/Ti-800 and Au@SiO$_2$/Ti-800-NaOH was performed in the temperature range of 100 $°C$ to $400\,°C$, with a gas flow of 30 mL min$^{-1}$ of 10 vol.% H$_2$O, 2 vol.% CO in He (space velocity of 18 L g$_{cat.}$$^{-1}$ h$^{-1}$). The inlet and outlet gas compositions were analyzed online by a gas chromatograph (Agilent-GC6890N) equipped with a flame ionization detector (FID) and a thermal conductivity detector (TCD).

## Data availability

The data that support the findings of this study are available within the paper and its Supplementary Information, and all data are available from the authors on reasonable request.

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

## Acknowledgements

This paper is dedicated to the 70th anniversary of the Dalian Institute of Chemical Physics, Chinese Academy of Sciences. This work was supported by National Natural Science Foundation of China (21473186, 21606223, and 21776270). J.H. acknowledges the funding from "Transformational Technologies for Clean Energy and Demonstration", Strategic Priority Research Program of the Chinese Academy of Sciences (XDA21030900). B.Q. acknowledges the support form National Key Projects for Fundamental Research and Development of China (2016YFA0202801), "Strategic Priority Research Program" of the Chinese Academy of Sciences (XDB17020100) and DNL Cooperation Fund, CAS DNL180403. B.Z. acknowledges the support from National Natural Science Foundation of China (No.21761132025) and the Youth Innovation Promotion Association CAS (2015152). R.S. acknowledges the support from National Natural Science Foundation of China (No.21773288). Finally, we would like to present our sincere thanks to Dr. Na Ta for her kind help in AC- HAADF-STEM measurement.

## Author contributions

Y.Z. and J.Z. conceived and synthesized the catalysts and performed most of the reactions and characterizations. Y.Z. and J.Z. contributed equally to this work. B.Z., Y.N., and F.H. performed the electron-microscopy characterization. R.S. carried out the X-ray absorption fine structure and analysis; B.H. and L.S. helped with the catalytic performance tests and discussed the results. L.L. did the DRIFT experiments. K.S. did the theoretical calculations and analysis. J.H., B.Q., and K.S. wrote the manuscript. J.H., B.Q., and M.H. designed the study and supervised the project.

## Competing interests

The authors declare no competing interests.
