## [Peer Review File · Nature Communications]

Reviewers' comments:

Reviewer #1 (Remarks to the Author):

This manuscript describes a new supported Au nanocatalyst that is stable above 800 °C. The catalyst, based on O-Au-O linkages stabilized by SiO₂ particles, appears to be extremely active for O₂ activation and resistant to sintering--even upon extensive thermal cycling.

The manuscript is very well written and the data provides a clear demonstration of a sinter-resistant Au catalyst toward CO oxidation. The work appears to be a significant step toward stabilization of nanoparticulate Au within a TiO₂ support. The results are certain to generate a great deal of interest and research into this new system. Therefore, publication is recommended after the authors consider a few minor points:

(1) The DFT calculations provide an adsorption free energy of O₂ on the Au/TiO₂ catalyst of 0.91 eV compared to 0.45 eV on the Au@SiO₂/TiO₂ catalyst: what are the relative enthalpic and entropic components for this energy? Why, from an electronic structure perspective, are they different?

(2) The computed energetics and associated discussion assumes that the reaction mechanism is unchanged for the two catalysts. The degree of thermodynamic rate control does not necessarily prove that the reaction does not change from an LH to an MvK type process. The authors should perform some initial labeling studies to show the overall mechanism is the same for the two systems under consideration. Could SiO₂ activate lattice oxygen from the titania?

(3) The new work should be placed within the context of a recent review article on this topic: Surface chemistry of Au/TiO₂: Thermally and photolytically activated reactions, Surface Science Reports 71 (1), 77-271.

Reviewer #2 (Remarks to the Author):

In this work, the authors developed a simple protocol to co-deposit Au and SiO₂ onto TiO₂, for use in Au-catalyzed CO oxidation and RWGS. The observations are definitely interesting, and it is compelling that the Au nanoparticles are clearly prevented from sintering at high temperatures as, happens without the SiO₂ shell. The authors do provide evidence via STEM_EDS that the SiO₂ shell is overlapping the Au, and that Si is elsewhere on the surface, but further details of the nature of this overcoat (the specifics of how it restructures at high T) may be based on an over-generous interpretation of the microscopy. Figure S8 appears to be incompletely labeled, which may be contributing to the problem. In that figure there is no discrimination between S8A, in which the particles look SiO₂ coated, and S8B, in which the particles look bare.

I also have a hard time understanding how the Au was 75% cationic in the case of Au@SiO₂/Ti-300, as per the XPS. This is inconsistent with the 3.5 nm nanoparticles observed in TEM. Are the particles reducing in the electron beam in the TEM, or is the XPS mis-calibrated or incorrectly curve fit? If the particles had not yet reduced by 300C for some reason, that would also help explain the poor catalytic performance of the Au@SiO₂/Ti-300 samples. Indeed, the low performance of that sample was never fully explained.

The authors primarily explain the improved rates on their new materials by resorting to DFT and a microkinetic model. The authors do not include any part of their model or how they go about determining the rate constants, or how it was fit to experimental data, except to claim an 8-order improvement in rates. The authors appear to be claiming that a 0.03 eV difference in the O₂ dissociation activation barrier is responsible for the activity differences, but I don't see that difference being significant either in terms of producing different rates or in the ability of the calculations to discriminate among transition states that similar in energy. The authors do estimate that O₂ is significantly less unstable on the mixed oxide surface, but it is not obvious how that leads directly to 8 orders of magnitude in rate estimated by their microkinetic model, since they include no details on the model.

Overall, I don't have great confidence in the authors ability to fit the observations to a microkinetic model, since the rates were typically compared at high conversion. I understand that it is convention in the field of low-T CO oxidation, but it doesn't allow for kinetic insights. The problem with this is evident when their Table S5 suggests only 2x drop in rate when the SiO₂ layer is removed, while their theory suggests it should be 8 orders of magnitude. I suspect this issue can be rectified with better discussion of how the various conclusions were reached.

Coming back to the DFT predictions, which are the sole method by which the improved rates are explained, the authors propose a very specific mixed TiO₂-SiO₂ support as being responsible for the

improved rates. That mixed oxide is not supported by experiment, and I fear that the authors may have created some sort of very defective TiO₂ by inserting a Si atom into the lattice. Overall, there is no clear evidence that their simulation matches reality.

Instead, I wonder if the SiO₂-coated Au particles are a red herring? Are there some even smaller gold particles on the surface that might be responsible? Would the authors find that the nanoparticle size distribution were different if they used a different technique to measure it (ie EXAFS) without e-beam exposure? Given that the Au/Ti-300 is still faster than the Au@SiO₂/Ti-800, the specific TiO₂/SiO₂ surface is clearly not responsible for all rate improvements. This reaction system is famous for having small changes in nanoparticle dispersion give a big effect. There is a clear improvement in rates and stability, but I am simply not sold on the explanation provided, as is necessary for publication in this journal.

Reviewer #3 (Remarks to the Author):

This is an interesting paper on stabilizing Au catalysts for selective oxidation. It can be published after consideration of the following issues:

(1) One of the methods to probe the surface Au species is to run DRIFTS measurement of CO on catalysts. The authors should run DRIFTS to verify their claims.

(2) Can the authors carry out EELS measurement of the interface between SiO₂-Au through their STEM or TEM experiments? This may help to probe the local electronegativity of their samples for substantiating their electronegativity claim.

(3) The SiO₂ over-coating of Au catalysts has been demonstrated with ALD of SiO₂ on Au-TiO₂. See: Ma, Z.; Brown, S.; Howe, J. Y.; Overbury, S. H.; Dai, S. Surface modification of Au/TiO₂ catalysts by SiO₂ via atomic layer deposition. *J. Phys. Chem. C* 2008, 112, 9448-9457. 10.1021/jp801484h. This paper needs to be cited.

(4) In addition, this overcoating strategy bears similarity to a double confinement strategy (Peng, H. G.; et. al., Confined Ultrathin Pd-Ce Nanowires with Outstanding Moisture and SO₂ Tolerance in Methane Combustion. *Angew. Chem.-Int. Edit.* 2018, 57, 8953-8957. 10.1002/anie.201803393). This paper should be cited.

Response to reviewers' comments

Reviewers' comments:

We thank all the reviewers for their nice comments and valuable suggestions which are helpful in improving our manuscript.

Reviewer #1 (Remarks to the Author):

This manuscript describes a new supported Au nanocatalyst that is stable above 800 °C. The catalyst, based on O-Au-O linkages stabilized by SiO₂ particles, appears to be extremely active for O₂ activation and resistant to sintering--even upon extensive thermal cycling.

The manuscript is very well written and the data provides a clear demonstration of a sinter-resistant Au catalyst toward CO oxidation. The work appears to be a significant step toward stabilization of nanoparticulate Au within a TiO₂ support. The results are certain to generate a great deal of interest and research into this new system. Therefore, publication is recommended after the authors consider a few minor points:

Response:

Thank you for your nice comments and good questions.

(1) The DFT calculations provide an adsorption free energy of O₂ on the Au/TiO₂ catalyst of 0.91 eV compared to 0.45 eV on the Au@SiO₂/TiO₂ catalyst: what are the relative enthalpic and entropic components for this energy? Why, from an electronic structure perspective, are they different?

Response:

The temperature is 273.15 K and the partial pressure of O₂ is 20 000 Pa, and the contribution of entropy is 0.53 eV of O₂ (including the transition entropy and rotation entropy). We ignored the contribution of vibration entropy because the vibrational partition functions between gaseous phase and adsorbed state is rather small under typical reaction conditions. The adsorbed energy (normally regarded as enthalpy

change) is 0.38 eV and -0.08 eV on Au/TiO₂ and Au@SiO₂/TiO₂ catalyst, respectively.

As we have mentioned in the text, the electronegativity of Si⁴⁺ is 17.10 which is higher than that of Ti⁴⁺ (13.86), indicating that the Si⁴⁺ has stronger ability to attract electrons than Ti⁴⁺. The calculated Bader charge of Au in Si-O-Au structure and in Ti-O-Au structure are +0.23 and +0.16, respectively, which indeed confirms that the Si⁴⁺ attracts more electrons from neighboring Au than Ti⁴⁺. In addition, as we have discussed in our previous work (ChemCatChem 2013, 5, 2217-2222), the HOMO orbital (the bands at Fermi level) is the anti-bonding of linear O-Au-O structure, which also can be found in the **Figure S22**. **Figure S22** shows the calculated partial density of states of adsorbed O₂ on Au/TiO₂ (upper) and Au@SiO₂/TiO₂ (bottom). It can be found that the bands at the range of -7.5~ -8.0 eV is the bonding orbital of linear O-Au-O structures, and the bands at the Fermi level (-0.2 ~ 0.2 eV) is the anti-bonding orbital of linear O-Au-O structures. Because the bands at Fermi level are the anti-bonding orbitals of O-Au-O structures, Si⁴⁺ attracts more electrons from the anti-bonding orbitals than Ti⁴⁺. **Figure S22f** is the enlarged view of partial density of states at Fermi level, and it shows that more electrons locate in the anti-bonding of O-Au-O in Au/TiO₂ than that of Au@SiO₂/TiO₂. In total, from an electronic structure perspective, the Si⁴⁺ attracts more electrons from anti-bonding orbitals of O-Au-O structure and therefore it strengthens the bond of O-Au-O. The newly performed EELS examination confirms more electron locate on Si ions which covered gold NPs. Therefore, these Si ions are partly reduced as shown in **Figure 3**. The shorter O-Au bonds in O-Au-O of Au@SiO₂/TiO₂ compared to those of Au/TiO₂ also confirms the stronger O₂ adsorption as show in **Figure S22**. As a result, the O₂ adsorption is strengthened by Si⁴⁺ and the adsorption of O₂ is stronger on Au@SiO₂/TiO₂ than Au/TiO₂.

Figure S22. The calculated partial density of states of adsorbed O_2 on Au/TiO_2 (upper) and $Au@SiO_2/TiO_2$ (bottom). The inserted figures a-e show the partial charge density of some important bands and figure f is enlarged view of partial density of states at Fermi level.

Figure 3. TEM (left) and EELS spectra (right) of $Au@SiO_2/Ti-800$.

(2) The computed energetics and associated discussion assumes that the reaction mechanism is unchanged for the two catalysts. The degree of thermodynamic rate control does not necessarily prove that the reaction does not change from an LH to an MvK type process. The authors should perform some initial labeling studies to show the overall mechanism is the same for the two systems under consideration. Could SiO_2 activate lattice oxygen from the titania?

Response:

The reviewer raised a very good question. We acknowledged that CO oxidation on supported Au catalysts can mainly follow two mechanisms, i.e., L-H and MvK mechanism. However, only on very easily reducible oxide such as uncalcined FeO_x supported Au catalyst the MvK will dominate at low temperatures (*J. Catal.* 2013, **299**, 90-100); after calcination at elevated temperature MvK mechanism will not be prominent (*J. Catal.* 2013, **299**, 90-100, *ACS Catal.* 2015, **5**, 3528-3539). For TiO₂ supported Au catalysts, the MvK mechanism only became prominent at reaction temperature > 80 °C (*J. Catal.* 2010, **276**, 292-305; *Angew. Chem. Int. Ed.* 2011, **50**, 10241-10245). However, our catalysts in this work show distinct activity even at ambient temperature which suggests it does not follow a MvK mechanism. In addition, our DFT calculation shows that the formation energy of oxygen vacancy on Au/Ti(Si)O₂ is 2.83 eV, slightly higher than that on Au/TiO₂ (2.75 eV). The higher oxygen-vacancy formation energy makes the activation barrier of lattice oxygen increase from 0.36 eV to 0.41 eV. Therefore, the doping Si does not promote but slightly hinders the activation of lattice O. We therefore didn't think it's necessary to take MvK mechanism into account. To convince the reviewer, we performed further experiments to demonstrate this.

A H₂-TPR examination of Au/Ti-300 and Au@SiO₂/Ti-800 samples were first performed. The results show that on Au/Ti-300 there is a small reduction peak centered at ~300 °C which should be ascribed to the reduction of TiO₂ surface oxygen, while on Au@SiO₂/Ti-800 no reduction was observed (**Figure S19**). The results are in good agreement with the DFT calculation that the doping of SiO₂ actually inhibits the activation of lattice oxygen. EPR measurements reveal that on Au/TiO₂-300 there is a small signal of Ti³⁺ (corresponding to O vacancy) while on Au@SiO₂/TiO₂-800 sample no signal was observed (**Figure S13**), further corroborating the TRP examination.

Figure S19. H₂-TPR of Au/Ti-300 and Au@SiO₂/Ti-800 samples.

Figure S13. EPR spectra of Au/Ti-300, Au@SiO₂/Ti-800. $g=1.975$ displays the signal of Ti³⁺.

A CO pulse reaction was also performed on Au/Ti-300 and Au@SiO₂/Ti-800 at 50 °C (Figure S20). On both catalysts CO₂ product was detected but the intensity on Au@SiO₂/Ti-800 is much lower, in good consistent with above characterization results. In addition, the negligible CO₂ product compared with the CO reactant suggests that the MvK mechanism does not dominate at low temperature on both catalysts.

All these experimental data together with theoretical calculation unambiguously demonstrated that the introduction of SiO₂ didn't improve the activation of TiO₂ lattice oxygen.

Figure S20. CO pulse reactions of Au/Ti-300 (left) and Au@SiO₂/Ti-800 (right) at 50 °C.

(3) The new work should be placed within the context of a recent review article on this topic: Surface chemistry of Au/TiO₂: Thermally and photolytically activated reactions, Surface Science Reports 71 (1), 77-271.

Response:

Thank you for your good suggestion, we have discussed the mentioned nice work in our revision.

Reviewer #2 (Remarks to the Author):

In this work, the authors developed a simple protocol to co-deposit Au and SiO₂ onto TiO₂, for use in Au-catalyzed CO oxidation and RWGS. The observations are definitely interesting, and it is compelling that the Au nanoparticles are clearly prevented from sintering at high temperatures as, happens without the SiO₂ shell. The authors do provide evidence via STEM-EDS that the SiO₂ shell is overlapping the Au, and that Si is elsewhere on the surface, but further details of the nature of this overcoat (the specifics of how it restructures at high T) may be based on an over-generous interpretation of the microscopy. Figure S8 appears to be incompletely labeled, which may be contributing to the problem. In that figure there is no discrimination between S8A, in which the particles look SiO₂ coated, and S8B, in which the particles look bare.

Response:

Thank you very much for your nice comments and valuable questions and suggestions. For **Figure S8** all the images are for Au@SiO₂/Ti-800. We acknowledge that the coating layer on Au NPs is not very homogenous; in some cases the SiO₂ coating layer is thinner and in other cases it is thicker. In addition, in **Figure S8A** all images were obtained in a relatively higher magnification while in **Figure S8B** the images were obtained in a slightly lower magnification which makes the Au NPs looks seemingly bare. But in fact, the NPs in **Figure S8B** is just covered by a thinner SiO₂ coating layer. For example, when we enlarge one of the Au NPs in Figure S8B, the thin cover layer can be observed clearly (see below arrows indicated).

I also have a hard time understanding how the Au was 75% cationic in the case of Au@SiO₂/Ti-300, as per the XPS. This is inconsistent with the 3.5 nm nanoparticles observed in TEM. Are the particles reducing in the electron beam in the TEM, or is the XPS mis-calibrated or incorrectly curve fit? If the particles had not yet reduced by 300C for some reason, that would also help explain the poor catalytic performance of the Au@SiO₂/Ti-300 samples. Indeed, the low performance of that sample was never fully explained.

Response:

The reviewer raised a very good question. First of all, the XPS data has neither been mis-calibrated nor been incorrectly fitted. The latter can be confirmed by the almost exactly same shape of XPS spectrum to that of Au@SiO₂/Ti-fresh sample. Second, it should be noted that the 75% species are slightly positively charged rather than cationic ones. We apologize for incorrectly using the word of “cationic” in our original manuscript which was misleading. We have corrected this in the revision by changing to “positively charged”. The positively charged Au should originate from the strong interaction with SiO_x cover layer. However, whether this is responsible for the low activity of Au@SiO₂/Ti-300 remains unclear because whether positively charged Au are more active for CO oxidation is still under controversy [J. Am. Chem. Soc. 2004, 126, 2672-2673].

The authors primarily explain the improved rates on their new materials by resorting to DFT and a microkinetic model. The authors do not include any part of their model or how they go about determining the rate constants, or how it was fit to experimental data, except to claim an 8-order improvement in rates. The authors appear to be claiming that a 0.03 eV difference in the O₂ dissociation activation barrier is responsible for the activity differences, but I don't see that difference being significant either in terms of producing different rates or in the ability of the calculations to discriminate among transition states that similar in energy. The authors do estimate that O₂ is significantly less unstable on the mixed oxide surface, but it is not obvious how that leads directly to 8 orders of magnitude in rate estimated by their

microkinetic model, since they include no details on the model.

Response:

The micro-kinetic model was constructed by numerically solving the differential equations based on steady state approximation, and the details could be found elsewhere [J. Am. Chem. Soc. 2016, 138, 10467-10476; 2017, ACS Catalysis. 2017, 7 4281-4290; J. Phys. Chem. C. 2018, 122, 9523-9530]. In brief, the reaction rate constant (k_i) can be derived from the transition state theory (the Eyring equation).

$$k_r = \frac{k_B T}{h} \exp\left(\frac{-E_a}{k_B T}\right) \quad (1)$$

where k_B , h , T and E_a represent the Boltzmann constant, Planck constant, absolute temperature and the activation energy of the reaction. The rate constant of adsorption (k_{ads}) was calculated by the following:

$$k_{\text{ads}} = \frac{S \cdot P \cdot A}{\sqrt{2\pi m k_B T}} \quad (2)$$

where S , P , A and m are the sticking coefficient, partial pressure of the adsorbed species, the area of the adsorption site, and the molecular mass of adsorbed species respectively. The value of S is assumed to one for all the species. The rate constant of desorption was calculated by the expression,

$$k_{\text{des}} = \frac{k_B T^3}{h^3} \frac{A(2\pi m k_B)}{\sigma \theta_{\text{rot}}} \exp\left(\frac{-E_{\text{des}}}{k_B T}\right) \quad (3)$$

where σ , θ_{rot} and E_{des} are the symmetry number, the characteristic temperature for rotation of adsorbed species, and the activation energy of desorption. The steady state approximation is adopted, assuming that the coverage changes of surface species are zero. The degree of thermodynamic rate control $X_{\text{RC},i}$ of step i , which can qualitatively determine the significance of single step i in the overall reaction, is calculated as following,

$$X_{\text{RC},i} = \frac{k_i}{r} \left(\frac{\partial r}{\partial k_i} \right)_{k_{j \neq i}, K_i} = \left(\frac{\partial \ln r}{\partial \ln k_i} \right)_{k_{j \neq i}, K_i} \quad (4)$$

where the partial derivative is taken holding constant for the rate constant k_j ($j \neq i$) and equilibrium constant K_i .

It should be noted that the rate constant of adsorption (k_{ads}) and desorption (k_{des}) were calculated by the equilibrium method in the previous submitted manuscript, following the method in the literature [ACS Catalysis. 2017, 7 4281-4290;]. This method will overestimate the k_{ads} and k_{des} in most cases, therefore it overestimates the reaction rates especially for the reactions with high reaction rate. To obtain more reliable results, we use Hertz-Knudsen method, following the method in the literature [J. Am. Chem. Soc. 2016, 138, 10467-10476], the details are briefly described in the previous part. The calculated reaction rate for Au/TiO₂ and Au/TiO₂@SiO₂ are $3.01 \times 10^{-7} \text{ s}^{-1} \text{ site}^{-1}$ and $0.702 \text{ s}^{-1} \text{ site}^{-1}$. The calculated difference of reaction rate between Au/TiO₂ and Au/TiO₂@SiO₂ is 6 orders of magnitude by Hertz-Knudsen method. It should be noted that the difference of 0.03 eV in the O₂ dissociation activation barrier is NOT mainly responsible for 6-order improvement in rates. The significant difference in rates can be assigned to the difference of O₂ coverage on Au/TiO₂ and Au/TiO₂@SiO₂. The coverages of O₂ are 1.49×10^{-17} and 2.10×10^{-10} on Au/TiO₂ and Au/TiO₂@SiO₂, respectively. The difference of O₂ coverage is caused by the difference of adsorption energies. The adsorption energies of O₂ are 0.38 eV and -0.08 eV on Au/TiO₂ and Au/TiO₂@SiO₂, respectively. The difference of adsorption energy is derived from the different electronegativity of Si⁴⁺ and Ti⁴⁺.

As we have mentioned in the text, the electronegativity of Si⁴⁺ is 17.10 eV which is higher than that of Ti⁴⁺ (13.86 eV). It indicates that the Si⁴⁺ has stronger ability to attract electrons than Ti⁴⁺. The calculated Bader charge of Au in Si-O-Au structure and in Ti-O-Au structure is +0.23 and +0.16, respectively, confirming that the Si⁴⁺ attracts more electrons from neighboring Au than Ti⁴⁺. In addition, as we have discussed in our previous work (ChemCatChem 2013, 5, 2217-2222), the HOMO orbital (the bands at Fermi level) is the anti-bonding of linear O-Au-O structure, which also can be found in the **Figure S22**. **Figure S22** shows the calculated partial density of states of adsorbed O₂ on Au/TiO₂ (upper) and Au@SiO₂/TiO₂ (bottom). It can be found that the bands at the range of -7.5~ -8.0 eV is the bonding orbital of linear O-Au-O structures, and the bands at the Fermi level (-0.2 ~ 0.2 eV) is the anti-bonding orbital of linear O-Au-O structures. Because the bands at Fermi level are

the anti-bonding orbitals of O-Au-O structures, Si^{4+} attracts more electrons from the anti-bonding orbitals than Ti^{4+} . **Figure S22f** is the enlarged view of partial density of states at Fermi level, and it shows that more electrons locate in the anti-bonding of O-Au-O in Au/TiO₂ than that of Au@SiO₂/TiO₂. In total, from an electronic structure perspective, the Si^{4+} attracts more electrons from anti-bonding orbitals of O-Au-O structure and therefore it strengthens the bond of O-Au-O. The newly performed EELS examination confirms more electron locate on Si ions which covered gold NPs. Therefore, these Si ions are partly reduced as shown in **Figure 3**. The shorter O-Au bonds in O-Au-O of Au@SiO₂/TiO₂ compared to those of Au/TiO₂ also confirms the stronger O₂ adsorption as show in **Figure S22**. As a result, the O₂ adsorption is strengthened by Si^{4+} and the adsorption of O₂ is stronger on Au@SiO₂/TiO₂ than Au/TiO₂.

Figure S22. The calculated partial density of states of adsorbed O₂ on Au/TiO₂ (upper) and Au@SiO₂/TiO₂ (bottom). The inserted figures a-e show the partial charge density of some important bands and figure f is enlarged view of partial density of states at Fermi level.

Figure 3. TEM (left) and EELS spectra (right) of Au@SiO₂/Ti-800.

Overall, I don't have great confidence in the authors ability to fit the observations to a microkinetic model, since the rates were typically compared at high conversion. I understand that it is convention in the field of low-T CO oxidation, but it doesn't allow for kinetic insights. The problem with this is evident when their Table S5 suggests only 2x drop in rate when the SiO₂ layer is removed, while their theory suggests it should be 8 orders of magnitude. I suspect this issue can be rectified with better discussion of how the various conclusions were reached.

Response:

The reviewer raised a very good question. We would like to clarify first that all the reaction rates (TOF) were obtained in a differential mode where the CO conversions were kept at low conversions (<15%) with much higher space velocity (50 000-100 000 mL g⁻¹cat. h⁻¹) to avoid heat effect and diffusion effect. We apologize for not making this clear in our original manuscript and have now added the details of reaction rate and TOF measurement into the revised manuscript.

The difference between calculated reaction rates (the unit is s⁻¹ site⁻¹) in several orders of magnitude could not be directly linked to a large different reaction rate on real catalysts where the unit is s⁻¹ because the accurate determination of the active sites is extremely difficult in real catalyst. The DRIFT spectra of CO adsorption have clearly revealed that the deposition of SiO₂ can cover lots of Au surface sites (**Figure S10**). Therefore, after the removal of SiO₂ cover layer much more Au surface active

sites were exposed albeit their intrinsic activity is much lower. The much more active sites can to some extents offset the much lower activity, thus presenting an apparent low-degree CO conversion drop. We have added the discussion into our revised manuscript.

Coming back to the DFT predictions, which are the sole method by which the improved rates are explained, the authors propose a very specific mixed TiO₂-SiO₂ support as being responsible for the improved rates. That mixed oxide is not supported by experiment, and I fear that the authors may have created some sort of very defective TiO₂ by inserting a Si atom into the lattice. Overall, there is no clear evidence that their simulation matches reality.

Response:

The reviewer raised a very good question. First of all, the proposed model is based on our experimentally fabricated catalyst structure, i.e., Au bonded to SiO₂ which supported on TiO₂. Therefore, the key is to form suitable Au-SiO₂ interface and bonding rather than the formation of SiO₂-TiO₂ mixture. The reviewer's concern that the improved activity may actually originate from the formation of some sort of very defective TiO₂ is very similar to question 2 of reviewer #1. We have demonstrated, theoretically and experimentally, that it is not the case. Details see response to question 2 of reviewer #1. In addition, to further release the reviewer's concern, we have performed further experiment to exclusively rule out the formation of highly active Au-TiO₂ interface. Yates group recently demonstrated a new CO oxidation mechanism on Au/TiO₂ by low-temperature DRIFT spectroscopy (Science, 2011, 333(6043):736-739) which has been developed into a new method to examine the interfaces between the metal NPs and the TiO₂ support (Angew. Chem. Int. Ed. 2016, **55**, 2820-2824; Angew. Chem. Int. Ed. 2016, **55**, 10606-10611). We therefore performed a similar measurement at -150 °C and -100 °C, respectively. In both cases after the introduction of gas mixture of CO and air no CO₂ generation was observed at all, suggesting the inexistence of such a highly active Au-TiO₂ interface (**Figure S14**).

Figure S14. The in-situ DRIFT spectra of CO adsorption at different temperature on Au/SiO₂/Ti-800 after introducing 500 Pa gas mixture of 1% CO and air.

Instead, I wonder if the SiO₂-coated Au particles are a red herring? Are there some even smaller gold particles on the surface that might be responsible? Would the authors find that the nanoparticle size distribution were different if they used a different technique to measure it (ie EXAFS) without e-beam exposure? Given that the Au/Ti-300 is still faster than the Au@SiO₂/Ti-800, the specific TiO₂/SiO₂ surface is clearly not responsible for all rate improvements. This reaction system is famous for having small changes in nanoparticle dispersion give a big effect. There is a clear improvement in rates and stability, but I am simply not sold on the explanation provided, as is necessary for publication in this journal.

Response:

This is a good question. We agree with the reviewer that the size of Au NPs would have significant effect on the activity of CO oxidation. That's the reason why we make a lot of control experiments to exclude the size effect such as comparing the activity of Au@SiO₂/Ti-800 before and after the removal of SiO₂ coating layer, and comparing with the Au/TiO₂-600. Actually, the purpose of control experiment of leaching SiO₂ layer is to exclude the size effect where a significant size change is not supposed to happen. We think these series experiments have already excluded the size

effect. However, to further convince the reviewer, we have performed more experiments. First, we have examined the samples with aberration corrected (AC) scan transmission electron microscopy (STEM) with a guaranteed resolution of 0.08 nm where all small nanoparticles and even single atoms can be observed. The results show that that no very small Au NPs were observed (**Figure S11**). In addition, according to the reviewer's suggestion, we have also performed EXAFS measurements and the results show that Au@SiO₂/Ti-800 do have much larger Au NPs (higher coordination number of Au-Au) compared with Au/Ti-300 (**Table S6**).

Figure S11. AC-HADDF-STEM images of Au@SiO₂/Ti-800.

sample	shell	R (Å)	CN	ΔE_0 (eV)	σ^2 (Å ²)
Au@SiO ₂ /Ti-fresh	Au-O	1.95 ± 0.01	3.2 ± 0.5	9.2 ± 0.9	
	Au-Au	2.88 ± 0.02	2.2 ± 0.7		
Au@SiO ₂ /Ti-300	Au-Au	2.86 ± 0.01	8.4 ± 1.3	7.2 ± 0.9	0.004 ± 0.002(O)
Au@SiO ₂ /Ti-800	Au-Au	2.86 ± 0.01	9.2 ± 1.8	7.2 ± 0.9	
Au/Ti-fresh	Au-O	1.95 ± 0.02	3.4 ± 1.0	9.2 ± 0.9	0.008 ± 0.001(Au)
	Au-Au	2.90 ± 0.06	3.0 ± 2.6		
Au/Ti-300	Au-Au	2.84 ± 0.01	7.4 ± 1.4	7.2 ± 0.9	
Au/Ti-800	Au-Au	2.87 ± 0.01	11.8 ± 2.2	7.2 ± 0.9	

Table S6. Au L₃-edge EXAFS fitting results (R : distance; CN : coordination number; σ^2 : Debye-Waller factor; ΔE_0 : inner potential correction) of different gold catalysts.

Reviewer #3 (Remarks to the Author):

This is an interesting paper on stabilizing Au catalysts for selective oxidation. It can be published after consideration of the following issues:

Thank you for your nice comments and helpful suggestions which make the work more publishable.

(1) One of the methods to probe the surface Au species is to run DRIFTS measurement of CO on catalysts. The authors should run DRIFTS to verify their claims.

Response:

Thank you for your valuable suggestion! According to your suggestion we have performed CO-DRIFT measurements at -150 °C where CO adsorption on TiO₂ can be detected as well. The results are presented in **Figure S10**. As expected, Au/Ti-300 sample has highest CO adsorption on both Au and TiO₂ surface. However, on Au/Ti-800 sample the CO adsorption on TiO₂ sites decreased significantly due to the decrease of surface area, while that on Au sites disappeared totally probably due to the very weak adsorption of CO on large Au particles (Angew. Chem. Int. Ed. 2016, 55, 10606-10611). On the other hand, on Au@SiO₂/Ti-300 CO adsorption on both TiO₂

and Au decreased dramatically, evidencing the coating effect of SiO₂. Interestingly, after calcined at 800 °C, the CO adsorption on TiO₂ increased while that on Au sites decreased further. The former is in well agreement with the shrink of SiO₂ layer during calcination and the latter is due to the aggregation of gold particles.

Figure S10. DRIFT spectra of CO adsorption on various catalysts at -150 °C.

(2) Can the authors carry out EELS measurement of the interface between SiO₂-Au through their STEM or TEM experiments? This may help to probe the local electronegativity of their samples for substantiating their electronegativity claim.

Response:

The reviewer proposed a very good suggestion, thank you! According to your suggestion we have performed EELS analysis of the interface between Au and SiO₂ (Figure 3). The results suggested that on TiO₂ support Si exhibited typical Si-O₄ tetrahedral structure, suggesting that Si existed as SiO₂. However, on Au NPs, Si exhibited more metal properties, suggesting that the electron transfer from Au to SiO₂ occurred. The result is in good consistent with the XPS result and further confirms the strong interaction between Au and SiO₂.

We have added all the new data and corresponding discussion into the revised manuscript.

Figure 3. TEM (left) and EELS spectra (right) of Au@SiO₂/Ti-800.

(3) The SiO₂ over-coating of Au catalysts has been demonstrated with ALD of SiO₂ on Au-TiO₂. See: Ma, Z.; Brown, S.; Howe, J. Y.; Overbury, S. H.; Dai, S. Surface modification of Au/TiO₂ catalysts by SiO₂ via atomic layer deposition. *J. Phys. Chem. C* 2008, 112, 9448-9457. 10.1021/jp801484h. This paper needs to be cited.

Response:

Thank you so much for kindly bring this nice work. We have discussed and cited this good work in our revision.

(4) In addition, this overcoating strategy bears similarity to a double confinement strategy (Peng, H. G.; et. al., Confined Ultrathin Pd-Ce Nanowires with Outstanding Moisture and SO₂ Tolerance in Methane Combustion. *Angew. Chem.-Int. Edit.* 2018, 57, 8953-8957. 10.1002/anie.201803393). This paper should be cited.

Response:

Thank you so much for kindly bring this nice work. We have cited it in our revision.

Reviewers' comments:

Reviewer #1 (Remarks to the Author):

The authors have adequately and carefully addressed the major concerns associated with the original manuscript. This reviewer judges the manuscript to be of high quality and the work provides an important contribution to the field. Therefore publication is recommended without further major changes.

Reviewer #2 (Remarks to the Author):

The revision added important additional experiments including some very useful spectroscopy.

I stand by my initial assessment that it is certainly interesting and valuable that the authors have found that Au and Si co-deposition results in a catalyst that retains activity (indeed, improves its reactivity) after a high temperature treatment. However, as a catalysis study it is lacking.

The authors create a computational model of the Au/SiO₂/TiO₂ interface whose specific structure has no obvious connection to reality. From this they deduce a many orders of magnitude improvement in per-site reactivity that must, coincidentally, also result in a many orders of magnitude decrease in the number of sites, in order to explain the very moderate changes in overall rate. These proposed changes in the active site should be accompanied by changes in apparent activation energies, but no such study was carried out.

The authors also ascribe a lot of weight to the electronegativity of the Si atom. No similar enhancements have been seen from simple Au/SiO₂ or Au deposited on titania-silica, yet it would seem to follow from the author's arguments. Overall, the authors seem to be far too quick to dismiss more mundane reasons for the activity, such as small differences in the particle size distribution, amount of residual halide, or small differences in the O₂ activation at the mixed oxide interface.

Again, this is an interesting material, and the encapsulation route is clever, if not necessarily tunable. The structure at the Au/SiO₂ interface seems to be far from resolved (could a surface layer of gold silicide account for the EELS and XPS?), but merits further study. Ultimately however, there is no hard evidence that the catalytic activity in these materials arises from anything more than physically encapsulated Au/TiO₂.

Reviewer #2 (Remarks to the Author):

The revision added important additional experiments including some very useful spectroscopy.

Response:

We sincerely thank all the reviewers for their good questions and constructive suggestions which are very helpful in improving our work.

I stand by my initial assessment that it is certainly interesting and valuable that the authors have found that Au and Si co-deposition results in a catalyst that retains activity (indeed, improves its reactivity) after a high temperature treatment. However, as a catalysis study it is lacking.

Response:

While we totally agree with your initial assessment, we have to respectively disagree with the comment that “as a catalysis study this work is lacking”. Actually in this work we have not only demonstrated the excellent performance of our newly developed catalyst, but also performed a relatively comprehensive study to reveal the underlying reasons, which has been recognized by the other two reviewers. So we don't think this work is much lacking as a catalysis study.

The authors create a computational model of the Au/SiO₂/TiO₂ interface whose specific structure has no obvious connection to reality. From this they deduce a many orders of magnitude improvement in per-site reactivity that must, coincidentally, also result in a many orders of magnitude decrease in the number of sites, in order to explain the very moderate changes in overall rate. These proposed changes in the active site should be accompanied by changes in apparent activation energies, but no such study was carried out.

Response:

We fully understand the reviewer's concern. However, we would like to stress first that in heterogeneous catalysis it is extremely difficult, if not impossible, to create a theoretical model that is exactly same to the real catalyst counterpart due to the complexity of heterogeneous catalysts, including inhomogeneous size distribution, metal-support interaction, electron effect, geometric effect etc. Therefore, the value obtained by theoretical calculation can barely be exactly same to the one obtained by experimental measurement. Even for the single-atom catalyst (SAC), which might be the simplest heterogeneous catalyst and has been regarded to be promising in narrowing the gap between theoretical calculation and experiment (*Acc. Chem. Res.*, 2013, **46**, 1740-1748), it is hard to directly compare the calculated and measured results. For example, we recently reported a significant water promoting effect on CO oxidation on Au₁/CeO₂ single-atom catalyst (*Nature Communications* **2019**, *10* (1), 3824) where the calculated rate increment by the presence of water is about 7 orders of magnitude ($1.8 \times 10^4 \text{ s}^{-1}$ v.s. $1.73 \times 10^{11} \text{ s}^{-1}$) but the measured rate increment is only about 2 orders of magnitude. Therefore, our opinion is that we can compare the theoretical calculation and experimental measurement results to evaluate their trend, but cannot directly compare the exact values which are scarcely the same.

But we totally agree with you that the changes in active site nature will result in changes in apparent activation energies. We therefore calculated and measured the apparent activation energies which are totally different before and after removal of SiO₂ by NaOH (see below which has been added into the revised manuscript), unambiguously demonstrating the different nature of the active sites (**Figure S18**).

Figure S18. Arrhenius plots of (A) the experimental reaction rate $\ln(\text{TOF})$ vs $1/T$ for $\text{Au}@/\text{SiO}_2/\text{Ti}-800$ and $\text{Au}@/\text{SiO}_2/\text{Ti}-800\text{-NaOH}$ and (B) the theoretical reaction rate $\ln(r)$ vs $1/T$ for Au/TiO_2 and $\text{Au}@/\text{SiO}_2/\text{TiO}_2$.

The authors also ascribe a lot of weight to the electronegativity of the Si atom. No similar enhancements have been seen from simple Au/SiO_2 or Au deposited on titania-silica, yet it would seem to follow from the author's arguments. Overall, the authors seem to be far too quick to dismiss more mundane reasons for the activity, such as small differences in the particle size distribution, amount of residual halide, or small differences in the O₂ activation at the mixed oxide interface.

Response:

This is a good question. We had been thinking the same question for long time that since Si has high electronegativity and the formed $\text{Au}-\text{SiO}_2$ interface can activate O₂ more effectively, why no one has previously discovered this. Up to now we still don't have a definite answer yet. But base on the reference research and analysis we tend to believe that it is related to the different catalyst preparation methods used in this work from those previously reported. In previous studies, the Au and SiO₂ were usually loaded at different steps wherein a suitable $\text{Au}-\text{SiO}_2$ interface with strong interaction is often hard to obtain. For instance, Au/SiO_2 catalysts were usually prepared by (improved) impregnation method or deposition precipitation (DP) method where the Au precursors were post-deposition on SiO₂ powder/nanoparticles. Similarly, the common titania-silica supported Au catalysts were often prepared by post-coating of SiO₂ onto Au nanoparticles supported on TiO₂ or depositing Au precursors onto TiO₂-SiO₂ composites which is similar to the deposition of Au onto SiO₂. In all these cases, an atomic level mixture of Si and Au is impossible thus a strong interaction between Au and SiO₂ is less possible. However, in our case, the precursors of Au and Si were co-deposited on TiO₂ support where an atomic level of mixture and interaction of Au and Si is possible. Thus during the post-calcination process, although

segregation and sintering of Au occurs, a strong interaction between Au and SiO₂ may be kept, forming a special active sites for the O₂ activation and CO oxidation.

In addition, we have to respectively disagree with the reviewer's comment saying that we are far too quick to dismiss more mundane reasons for the activity. i) The strong size effect in supported Au catalysts has been well-known and this is the reason why we and other researchers focus on the development of sintering resistant Au nanocatalyst to keep Au nanoparticles small. So the first concern we care about is the size of Au NPs. We had expected that with the developed method in this work we can obtain very small Au NPs with high activity even after high temperature (800 °C) calcination. Unfortunately however, detailed characterizations (HRTEM, ac-STEM, EXAFS) and detailed studies (a series control experiments including the removal of SiO₂, the calcination at 600 °C to obtain a similar Au size etc., see our last response to reviewer) have revealed that this is not the case: While the size of Au NPs is much smaller than that of Au/Ti-800, it is certainly no smaller than those reported in previous literatures with similar heat-treatment [*Angew. Chem. Int. Ed.* 2016, **55**, 10606–10611; *ACS Catal.* 2015, **5**, 1078–1086; *J. Am. Chem. Soc.* 2016, **138**, 16130–16139]. ii) As to the effect of halide (chloride), we agree with you that it is also well-known that the halide has significant effect on the activity of Au [*Catal. Rev.* 1999, **41**, 319-388]. However, this effect is only pronounced at preparation at low pH value condition where the chloride is hard to remove. When the Au was deposited on support at a high pH value condition the chloride can be easily removed by water washing. After about 30 years' extensive studies, it has now become a common sense to remove chloride during preparation of supported Au catalysts for CO oxidation and the relatively completely removal of chloride has not been a serious issue anymore. To convince the reviewer, we examined the content of chloride in various catalysts by ion chromatography, **Table S7**. It shows that before and after NaOH treatment the chloride content is no higher than 3 ppm, only about 0.03% of that of Au. Of more importance, the Au@SiO₂/Ti-800-NaOH sample with lower chloride concentration has lower activity. We hence think that the high activity of Au@SiO₂/Ti-800 is not due to a low concentration of halide. iii) The improvement in O₂ activation on mixed oxides generally needs the mixing of oxides having different valences to generate vacancies (*Chem Rev*, 2013, **113**, 4391-4427). However, this is not the case for SiO₂ and TiO₂ as Ti and Si in either their own oxides or the mixed oxide have same valence. In fact another reviewer had a similar question and we had experimentally demonstrated that no extra O₂ activation ability was obtained on the Au/ST-800 samples (see our last response).

Again, this is an interesting material, and the encapsulation route is clever, if not necessarily tunable. The structure at the Au/SiO₂ interface seems to be far from resolved (could a surface layer of gold silicide account for the EELS and XPS?), but merits further study. Ultimately however, there is no hard evidence that the catalytic activity in these materials arises from anything more than physically encapsulated Au/TiO₂.

Response:

This is a very good question. When we first got the EELS data we had considered the possibility of forming gold silicide composites (*RSC Adv.*, 2015, **5**, 101726–101731; *Nature*, 2006, **440**, 69–

71). However, both XRD and EXAFS characterization have not revealed the presence of such species. And ascribing the high activity to gold silicide is an unusual viewpoint thus needs unusually strong evidence which is still lacking so far. Based on above and for the sake of safety, we preferred not to propose the gold silicide composites as active sites without solid evidence. We do agree with the reviewer that this point merits further study which is, however, beyond the scope of this work.

Table S7. The chlorine ion content and the ratio of chlorine ion and Au for different catalysts.

Sample	Cl ⁻ content (ppm)	Cl ⁻ /Au ratio
Au@SiO ₂ /Ti-800	3	0.0003
Au@SiO ₂ /Ti-800-NaOH	<3	<0.0003

REVIEWERS' COMMENTS:

Reviewer #2 (Remarks to the Author):

I appreciate that the authors have taken the extra step to calculate experimental barriers over the different materials. This provides far more confidence in the results than did the prior, single-point rate measurements. I am in favor of manuscript acceptance at this point.